# Active site alanine mutations convert deubiquitinases into high-affinity ubiquitin-binding proteins

Marie E Morrow[1], Michael T Morgan[1], Marcello Clerici[2,†], Katerina Growkova[3], Ming Yan[1], David Komander[4] (iD), Titia K Sixma[2], Michal Simicek[3,4] & Cynthia Wolberger[1,*] (iD)

## Abstract

A common strategy for exploring the biological roles of deubiquitinating enzymes (DUBs) in different pathways is to study the effects of replacing the wild-type DUB with a catalytically inactive mutant in cells. We report here that a commonly studied DUB mutation, in which the catalytic cysteine is replaced with alanine, can dramatically increase the affinity of some DUBs for ubiquitin. Overexpression of these tight-binding mutants thus has the potential to sequester cellular pools of monoubiquitin and ubiquitin chains. As a result, cells expressing these mutants may display unpredictable dominant negative physiological effects that are not related to loss of DUB activity. The structure of the SAGA DUB module bound to free ubiquitin reveals the structural basis for the 30-fold higher affinity of Ubp8$^{C146A}$ for ubiquitin. We show that an alternative option, substituting the active site cysteine with arginine, can inactivate DUBs while also decreasing the affinity for ubiquitin.

Keywords deubiquitinating enzyme; polyubiquitin; ubiquitin binding
Subject Categories Post-translational Modifications, Proteolysis & Proteomics; Structural Biology

## Introduction

Deubiquitinating enzymes (DUBs) play fundamental roles in ubiquitin signaling through their ability to remove ubiquitin from target proteins and disassemble polyubiquitin chains [1]. These enzymes cleave the isopeptide linkage between the C-terminus of ubiquitin and the substrate lysine or, in some cases, the peptide bond between ubiquitin and a substrate protein N-terminus. The human genome encodes more than 90 DUBs [2,3], which can be grouped into families based on their fold: ubiquitin-specific protease (USP), ubiquitin carboxyl-terminal hydrolase (UCH), ovarian tumor family (OTU), Machado–Joseph domain (MJD) family, and JAMM/MPN domain (JAMM), as well as the recently discovered MINDY and ZUFSP families [4–6]. Studies of deletions as well as disease-causing mutations have revealed specific functions for individual DUBs in biological processes including proteasomal degradation, protein trafficking, transcription, DNA repair, infection, and inflammation [7]. The involvement of DUBs in a variety of oncogenic [8], inflammation, and neurodegenerative pathways [9] has made these enzymes attractive targets for drug discovery [10].

A common approach to determining the role of a particular DUB in cellular pathways is to knock down expression of the endogenous DUB and express a catalytically inactive version of the enzyme [11–13]. With the exception of the JAMM domain family, which are metalloproteases, all other DUBs are cysteine proteases with a papain-like active site in which the catalytic cysteine is activated by an adjacent histidine [14]. Cysteine protease DUBs are typically inactivated by substituting the active site cysteine with another residue. Resulting changes in substrate ubiquitination or downstream signaling pathways in cells expressing the mutant DUB are generally assumed to be due to the absence of deubiquitinating activity, with the notable exceptions of OTUB1, which inhibits E2 enzymes by a mechanism independent of catalytic activity [15–17] and OTUD4, which serves as a scaffold for USP enzymes [18]. Whereas serine is the most conservative substitution for the active site cysteine, alanine substitutions are often used to avoid the possibility that mutants containing a serine substitution may retain some hydrolase activity.

We report here that some active site cysteine-to-alanine substitutions can dramatically increase the affinity of DUBs for either free ubiquitin or polyubiquitin chains. This increase in affinity can confound the interpretation of cell-based experiments, since the mutant DUB is not only incapable of cleaving ubiquitin from substrates but has gained the ability to sequester free ubiquitin and

1 Department of Biophysics and Biophysical Chemistry, Johns Hopkins University School of Medicine, Baltimore, MD, USA
2 Division of Biochemistry and Oncode Institute, Netherlands Cancer Institute, Amsterdam, The Netherlands
3 Faculty of Medicine, University of Ostrava, Ostrava, Czech Republic
4 Medical Research Council Laboratory of Molecular Biology, Cambridge, UK
*Corresponding author. Tel: +1 410 955 0728; E-mail: cwolberg@jhmi.edu
†Present address: Department of Biochemistry, University of Zurich, Zurich, Switzerland

polyubiquitin chains. Altering levels of free ubiquitin has been shown to give rise to off-target effects [19]. In addition, these mutant DUBs may stably associate with (poly)ubiquitinated substrates and thereby protect ubiquitin chains from cleavage by other DUBs or prevent interaction with ubiquitin receptors. The effects of such tight-binding DUB mutants thus have the potential to confuse interpretation because of the gain of tight ubiquitin-binding function.

We show here that mutating the active site cysteine of human USP4 and yeast Ubp8 to alanine increases the affinity of the DUB for mono- or diubiquitin by 10–150-fold. A similar effect of alanine substitution was previously found for the OTU enzymes, Cezanne [20] and OTULIN [21]. The structure of the heterotetrameric SAGA DUB module containing Ubp8$^{C146A}$ bound to free ubiquitin reveals the molecular basis for the increased affinity of monoubiquitin for the mutant enzyme. The alanine substitution alleviates steric hindrance by the active site cysteine sulfhydryl, allowing the C-terminal carboxylate of ubiquitin to form additional hydrogen bonds in the enzyme active site and thus accounting for the high affinity of the mutant enzyme for free ubiquitin. We show that substituting the active site cysteine with arginine in representative USP and OTU DUBs inactivates the enzymes while also disrupting binding to ubiquitin, generating an inert DUB. Based on these findings, we strongly recommend that cell-based and *in vivo* studies of DUBs avoid the use of active site alanine substitutions and to instead utilize substitutions such as arginine that ablate both enzymatic activity and ubiquitin binding.

# Results and Discussion

## Mutation of active site cysteine to alanine increases affinity of the SAGA DUB module for ubiquitin

The yeast SAGA complex is a transcriptional coactivator that is involved in transcription of all RNA polymerase II genes [22,23]. Among the SAGA activities are the removal of monoubiquitin from histone H2B, which promotes transcription initiation and elongation [24]. The deubiquitinating activity of SAGA resides in a four-protein complex known as the DUB module, which comprises the USP family catalytic subunit, Ubp8, as well as Sgf11, Sus1, and the N-terminal ~100 residues of Sgf73 [25,26]. Structural studies of the DUB module complexed with ubiquitin aldehyde [27] and with ubiquitinated nucleosomes [28] have revealed the overall organization of the DUB module and how it interacts with substrate. In addition to its ability to deubiquitinate histone H2B, the DUB module can also cleave a variety of ubiquitin substrates *in vitro* including ubiquitin-AMC and K48-linked diubiquitin [29]. The affinity of the DUB module for ubiquitinated nucleosome has been estimated at around 2 μM [28] and the $K_M$ for the model substrate, ubiquitin-AMC, has been estimated at 24 μM [29]; however, neither the $K_M$ nor binding affinity for other substrates is known.

In order to measure the affinity of the DUB module for other substrates using binding assays, we expressed and purified catalytically inactive versions of the DUB module containing Ubp8 with its active site cysteine, C146, substituted with either serine (C146S) or alanine (C146A). The absence of catalytic activity for both mutants was first verified in a ubiquitin-AMC cleavage assay (Fig EV1). We

measured the affinity of both mutant complexes for K48-linked diubiquitin using isothermal titration calorimetry (ITC; Fig 1E and F). Whereas DUB module containing Ubp8$^{C146S}$ bound to K48 diubiquitin with a $K_d$ of 4.6 μM, DUB module containing Ubp8$^{C146A}$ bound to K48-linked diubiquitin with a $K_d$ of 0.47 μM, representing 10-fold tighter binding. We also measured the affinity of the reaction product, monoubiquitin, to DUB module containing either wild-type or mutant Ubp8 (Fig 1A–C). Whereas DUB module containing wild-type Ubp8 or Ubp8$^{C146S}$ bound ubiquitin with a $K_d$ of 13.9 μM and 12.8 μM, respectively, the Ubp8$^{C146A}$ mutant bound ~30-fold more tightly to monoubiquitin with a $K_d$ of 0.43 μM.

## A Ubp8 C146A substitution enables hydrogen bonding with the ubiquitin C-terminus

To determine the structural basis for the marked increase in affinity for free ubiquitin when the active site cysteine is substituted with alanine, we solved the crystal structure of the SAGA DUB module containing Ubp8$^{C146A}$ bound to free ubiquitin at a resolution of 2.1 Å (Table 1 and Fig 2A). The overall fold and contacts with ubiquitin are virtually identical to those found in the structure of the wild-type enzyme bound to ubiquitin aldehyde, superimposing all atoms with an RMSD of 0.58 Å [27]. The active site of the C146A mutant is virtually identical to that in the wild-type apoenzyme [29], with no significant reordering of residues (Fig 2B) [29]. In the Ubp8$^{C146A}$ complex with free ubiquitin, the negatively charged carboxylate of the ubiquitin C-terminal Gly76 forms two hydrogen bonds with backbone amides from Ubp8 residues Thr145 and Ala146, as well as with active site residues, Asn141 and His427 (Figs 2B and EV2). Importantly, the observed position of the ubiquitin C-terminus would not be compatible with the presence of the wild-type active site residue, Cys146, since the sulfhydryl group would clash with the C-terminal residue of ubiquitin, Gly76 (Fig 2C). The multiple hydrogen bonding interactions observed between the C-terminal carboxylate of ubiquitin and Ubp8 can therefore only occur when the active Cys146 is replaced with the smaller alanine side chain, thus explaining the higher affinity of DUB module-Ubp8$^{C146A}$ for free ubiquitin as compared to the wild-type enzyme.

## Active site cysteine-to-alanine substitution increases the affinity USP4 for ubiquitin

Mutating the active site cysteine to alanine has a dramatic effect on the affinity of the human USP family DUB, USP4, for free ubiquitin. USP4 regulates a broad variety of cellular pathways, including TGF-β and NF-κB signaling as well as splicing [26,30,31]. The affinity of USP4 for free ubiquitin was measured by fluorescence polarization using ubiquitin labeled with an N-terminal fluorophore. As shown in Fig 3, the $K_d$ of ubiquitin for the wild-type enzyme is 92 ± 21 nM, whereas USP4 containing an alanine substituted for the active site cysteine, C311, binds ubiquitin with 0.60 ± 0.17 nM affinity, a ~150-fold difference [32]. The pre-steady-state kinetics of ubiquitin dissociation measured by fluorescence polarization in a stopped-flow device shows that the greater affinity of the USP4$^{C311A}$ mutant is due to a dramatic decrease in off-rate (Fig EV3A). Interestingly, ubiquitin dissociation has been shown to be promoted by USP4 N-terminal DUSP-Ubl domain and to regulate USP4 activity

[32]. The increase in affinity for the mutant enzyme is not unique to ubiquitin with a free C-terminus, as ubiquitin conjugated to either an 18-mer peptide or C-terminal fluorophore also binds with similar affinity to USP4[C311A] (Fig EV3C and D). Since the active sites of USP family DUBs are highly conserved, we speculate that the observed increase in binding affinity is due a relief of steric clash, as is the case for Ubp8.

**Substitution of the catalytic cysteine with arginine disrupts ubiquitin binding in USP and OTU class DUBs**

We sought to identify alternative active site mutations that would abrogate catalytic activity as well as reduce the affinity of the inactive DUB polyubiquitin chains or ubiquitinated substrates. We reasoned that substituting the active site cysteine with arginine could both inactivate

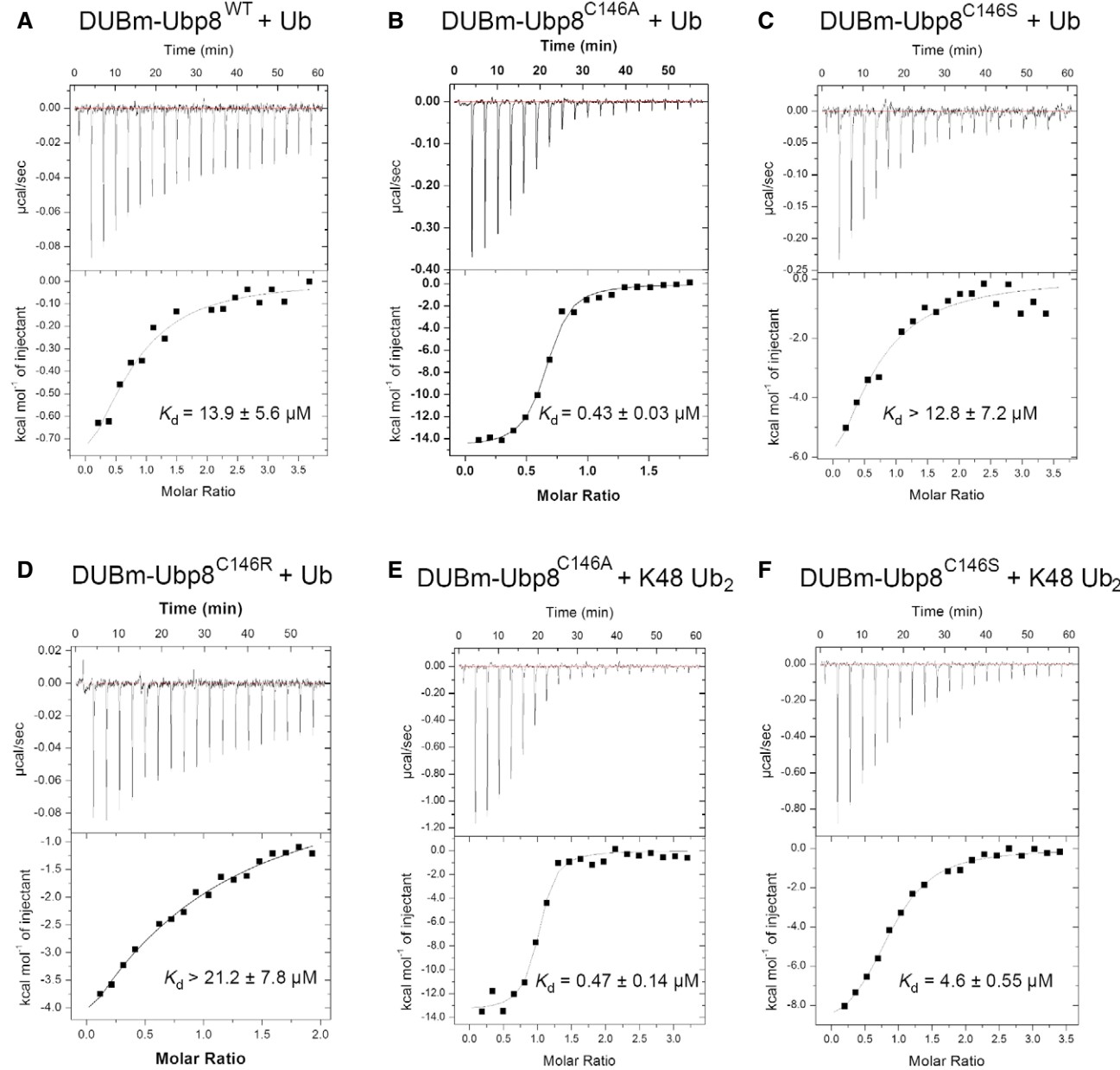

**Figure 1.   Isothermal titration calorimetry assays of SAGA DUB module binding to K48 diubiquitin or monoubiquitin.**

A   Binding of wild-type DUBm-Ubp8 to monoubiquitin.
B   Binding of DUBmUbp8[C146A] to monoubiquitin.
C   Binding of DUBm-Ubp8[C146S] to monoubiquitin.
D   Binding of DUBm-Ubp8[C146R] to monoubiquitin.
E   Binding of DUBm-Ubp8[C146A] to K48 diubiquitin.
F   Binding of DUBm-Ubp8[C146S] to K48 diubiquitin.

Data information: Error ranges for $K_d$ values were determined from nonlinear least squares fitting of the data to a one-site binding model.

**Table 1. X-ray crystallographic data and refinement statistics.**

| | |
|---|---|
| Wavelength (Å) | 0.979 |
| Resolution (Å) | 2.10 |
| Unique reflections | 54,195 |
| Redundancy | 5.7 (5.7) |
| Completeness (%) | 99.2 (99.7) |
| Average $I/\sigma$ (I) | 13.3 (3.0) |
| $R_{merge}$ | 0.093 (0.572) |
| $R_{meas}$ | 0.103 (0.630) |
| $R_{pim}$ | 0.043 (0.259) |
| CC1/2 | 0.998 (0.857) |
| CC* | 0.999 (0.961) |
| Refinement statistics | |
| Space group | $P2_12_12_1$ |
| Unit cell (Å) | $a = 78.8$, $b = 103.2$, $c = 112.8$ |
| Molecules per asymmetric unit | 1 |
| $R_{work}$ (%) | 20.1 |
| $R_{free}$ (%) | 24.9 |
| Rmsd bonds (Å) | 0.0198 |
| Rmsd angles (°) | 1.855 |
| Protein atoms | 6,302 |
| Zinc ions | 8 |
| Average B (Å$^2$) | 40.3 |

the enzyme and prevent ubiquitin binding because of the bulky nature of the side chain compared to cysteine. To test this hypothesis, we mutated the catalytic cysteine of Ubp8 to arginine and first verified that SAGA DUB module containing the mutant Ubp8$^{C146R}$ protein was inactive in a Ub-AMC cleavage assay (Fig EV1). The affinity of the DUB module containing Ubp8$^{C146R}$ for monoubiquitin as measured by ITC was comparable to that of the wild-type protein (Fig 1D).

Substitution of the active cysteine with arginine similarly reduces the affinity for polyubiquitin chains by the OTU family member, OTUD1, which is also a cysteine protease. This DUB preferentially cleaves K63-linked polyubiquitin chains [20], has recently been shown to regulate the nuclear localization and transcriptional coactivator activity of the YAP oncoprotein [33], represses metastasis by deubiquitinating SMAD7 during TGF-β signaling [34], and negatively regulates RIG-I-like receptor (RLR) signaling during viral infection by deubiquitinating Smurf1 [35]. We measured the affinity of catalytic mutants of OTUD1 for fluorescently labeled K63-linked diubiquitin using a fluorescence polarization assay (Fig 4A). While OTUD1 with an alanine substituted for the active site cysteine (OTUD1$^{C320A}$) binds K63-linked diubiquitin with a $K_d$ of ~40 μM, an arginine substitution, OTUD1$^{C320R}$, not only inactivated the enzyme but also completely abolished detectable binding to K63-linked diubiquitin (Fig 4A).

### Active site arginine substitutions can overcome artifacts of alanine substitution in cells

As mentioned above, active site cysteine-to-alanine substitutions that markedly increase DUB affinity for mono- or polyubiquitin may

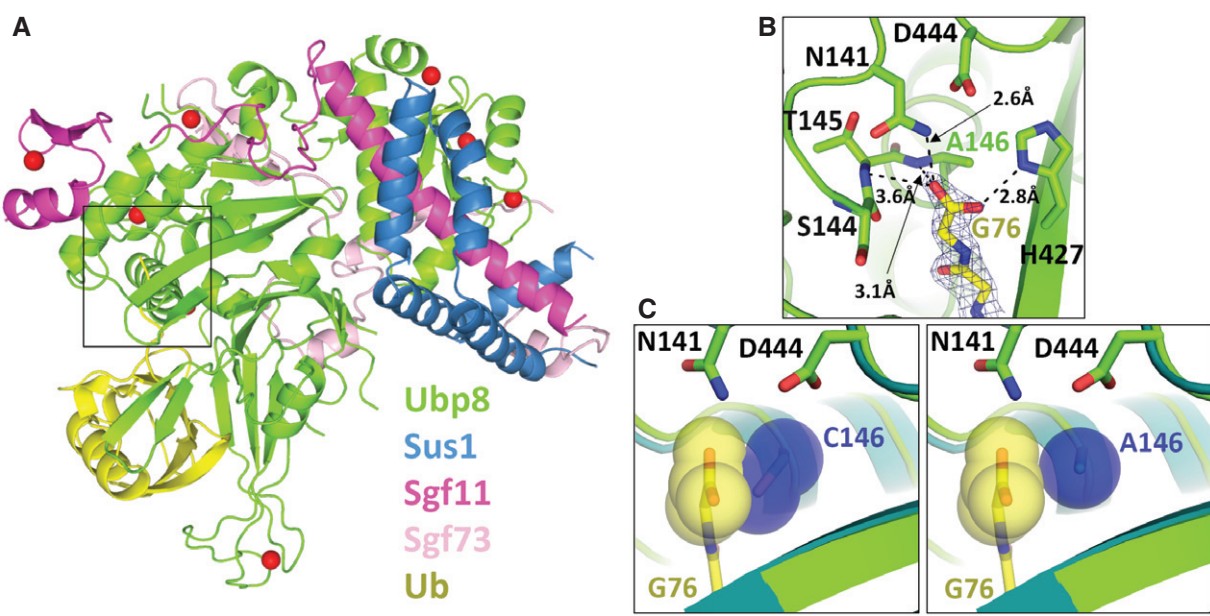

**Figure 2. X-ray crystal structure of SAGA DUB module mutant DUBm-Ubp8$^{C146A}$ bound to monoubiquitin.**

A Overall structure of complex showing Ubp8 (green) with ubiquitin (yellow) bound to the USP domain.
B Hydrogen bonding contacts between the C-terminal carboxylate of ubiquitin and Ubp8.
C In blue spheres, van der Waals radii of C146 and A146 in steric proximity of ubiquitin's C-terminal carboxylate (yellow). DUBm-Ubp8$^{WT}$ structure is shown in teal (PDB ID 3MHH) and DUBm-Ubp8$^{C146A}$ is shown in green.

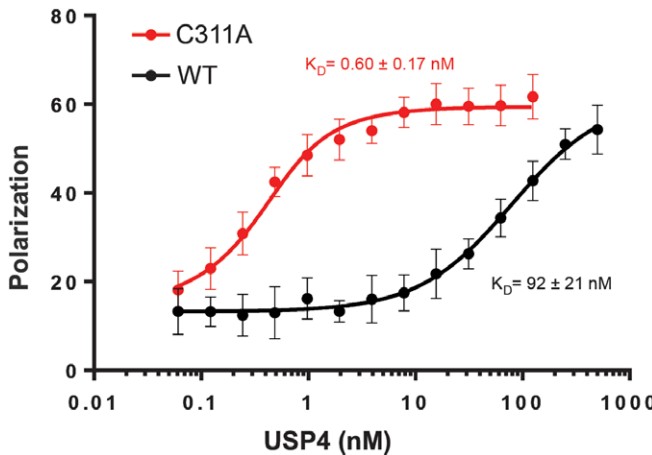

**Figure 3. Equilibrium binding of USP4 WT and C311A to TAMRA-labeled monoubiquitin.**

Binding was measured by fluorescence polarization using N-terminally TAMRA-labeled monoubiquitin. The dissociation constants for ubiquitin binding to USP4 WT and C311A are 92 ± 21 nM [32] and 0.60 ± 0.17 nM, respectively. Error bars are s.d. calculated on five measurements per point.

render these mutants less suitable for physiological studies. Since such cysteine-to-alanine mutants are essentially high-affinity ubiquitin-binding proteins, these DUB mutants have the potential to stabilize modified substrates and preferred chain types by protecting them from digestion by other DUBs or proteases. For example, when OTULIN C129A is expressed in cells, there is a dramatic accumulation of Met1-linked linear polyubiquitin chains that is not seen when a mutant that abrogates ubiquitin binding, L259E, is expressed [21]. We hypothesized that substituting the active site cysteine with arginine could be a general approach in cell-based studies to inactivating cysteine protease DUBs while also preventing high-affinity binding to polyubiquitin. To test this idea, we expressed

cysteine-to-alanine and cysteine-to-arginine mutants of two DUBs, OTUD1 and USP14, in cells and probed their effects on levels of polyubiquitin. HA-tagged wild-type OTUD1, OTUD1[C320A], or OTUD1[C320R] was expressed in HEK293 cells and whole cell lysates were analyzed by immunoblotting with an antibody specific for K63-polyubiquitin chains. As compared to cells expressing the wild-type protein, cells expressing OTUD1[C320A] had increased levels of K63-linked polyubiquitin (Fig 4B). By contrast, cells expressing OTUD1[C320R] did not show enriched levels of K63-linked chains (Fig 4B).

We also tested the effects of expressing wild-type and mutant USP14, one of the chain-trimming DUBs that bind to the 26S proteasome [36,37]. HA-tagged USP14 containing the wild-type active site cysteine, Cys114, and C114A and C114R mutants were co-expressed in HEK293 cells along with FLAG-PSMD4, a ubiquitin receptor within the proteasome [38]. Proteasome-bound ubiquitinated proteins were co-immunoprecipitated by FLAG-PSMD4 and probed for ubiquitin (Fig 5). Proteasomes with USP14 C114A bind more polyubiquitin chains than USP14 C114R and also retain increased levels of higher molecular weight chains that are unable to be trimmed compared to wild-type USP14 (Fig 5). Both of these results are consistent with the idea that the increase in polyubiquitin chains observed with the cysteine-to-alanine mutants is due to the ability of this mutant to bind to polyubiquitin chains and protect them from cleavage by other DUBs. Our results also validate the benefit of using a Cys to Arg substitution to generate a catalytically inactive DUB that will neither protect nor sequester polyubiquitin chains and ubiquitinated substrates.

### Implications for cell-based studies of cysteine protease DUBs

The surprisingly high affinity for ubiquitin exhibited by DUBs containing alanine substituted for the active site cysteine has important implications for cell-based assays in which catalytically inactive DUBs are expressed. We have found that cysteine-to-alanine

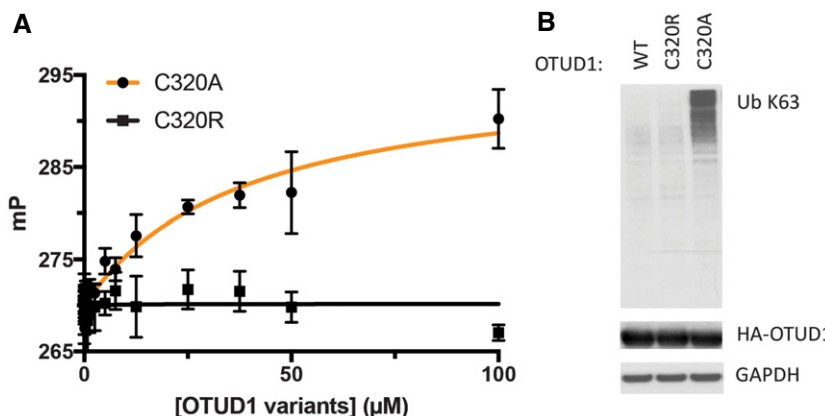

**Figure 4. Enhanced binding of OTUD1 C320A to K63 diubiquitin *in vitro* and K63 polyubiquitin chains in cells.**

A   Equilibrium binding of OTUD1 C320A and C320R to K63-linked diubiquitin was measured by fluorescence polarization using FlAsH-tagged K63-linked diubiquitin in which the proximal ubiquitin was fluorescently labeled. Error bars indicate s.d. and are based on three measurements per data point. One representative experiment of two is shown.

B   Whole cell lysates of HEK293 cells expressing HA-tagged OTUD1 WT, C320R, and C320A were immunoblotted with indicated antibodies. One representative experiment of three is shown.

**Figure 5.  Binding of USP14 C114A to ubiquitin chains in cells.**

Polyubiquitin chains were co-immunoprecipitated with FLAG-PSMD4, an ubiquitin receptor for the 26S proteasome, from cells expressing either USP14 wild-type, C114A, or C114R. One representative experiment of two is shown.

substitutions in the USP DUBs, Ubp8 and USP4, dramatically increase their affinity for monoubiquitin (Figs 1 and 3). Similarly, cysteine-to-alanine substitutions in the active sites of Ubp8 and the OTU class DUB, OTUD1, also increase DUB affinity for polyubiquitin (Figs 1 and 4A). The equilibrium dissociation constants of these mutant DUBs are significantly lower than cellular concentrations of their substrates, given the estimated concentration of free ubiquitin in the cell of 4–50 μM [39] and a concentration of polyubiquitin at a fraction of that [40]. The mutant DUBs are therefore expected to bind tightly to free ubiquitin and to polyubiquitin chains when expressed in cells. Particularly in experiments where mutant DUBs are overexpressed, there is the risk that cellular consequences ascribed to a lack of catalytic activity in a particular DUB may instead be due to the ability of the mutant DUB to protect polyubiquitin chains from cleavage or to a difference in free ubiquitin availability to ubiquitin-conjugating enzymes.

The effects of cysteine-to-alanine substitutions shown here for USP and OTU class DUBs are likely to extend to other cysteine protease DUBs. The UCH and MJD classes of cysteine protease DUBs have a conserved active site architecture similar to USP DUBs [14,41] and could thus form similar interactions with ubiquitin if the active site cysteine was substituted with alanine (Fig EV4A and B). Although the MINDY and ZUFSP DUBs share little homology with the other cysteine protease DUBs [4–6], an alanine substitution would similarly relieve steric clash in the active site, thus potentially increasing the DUB's affinity for ubiquitin (Fig EV4C). These observations support the recommendation that alanine substitutions should be avoided in cell-based and *in vivo* studies of cysteine protease DUBs.

We have presented an alternative to alanine substitutions that is equally effective in abrogating DUB activity while having the advantage of preventing ubiquitin binding. In binding assays of Ubp8 and OTUD1, we show that replacing the active site cysteine with arginine inactivates ubiquitin hydrolase activity while also rendering the enzyme incapable of binding ubiquitin detectably (Figs 1D and 4). We speculate that the ability of arginine substitutions to abolish

ubiquitin binding to OTU and USP catalytic domains may be explained by the ability of the arginine side chain to partially occupy the binding site for the C-terminal ubiquitin Gly-Gly in these DUBs. Previous data have indicated that the correct orientation of the ubiquitin C-terminal tail in the DUB S1 site is essential for efficient cleavage [42,43]. An arginine side chain may mimic these interactions in cis, to prevent these important substrate interactions. While we did not test other active site substitutions, it is expected that other amino acids with side chains that are significantly bulkier than cysteine would similarly block ubiquitin binding, in addition to inactivating the enzyme. However, care should be taken to avoid hydrophobic side chains that could cause the protein to aggregate, or side chains that are bulky or beta-branched that could interfere with proper protein folding due to steric clashes with the neighboring protein backbone. Since lysine can be ubiquitinated and is also subject to many other post-translational modifications, this substitution should also be avoided.

We recommend that all cell-based and *in vivo* studies of cysteine protease DUBs avoid alanine substitutions of the active site cysteine and instead utilize arginine substitutions to study effects of inactivating the enzyme. Although the arginine substitution was only tested here on DUBs from the USP and OTU class, it is likely that an arginine would similarly interfere with ubiquitin binding to members of the UCH, MJD, MINDY, and ZUFSP cysteine protease families. Ideally, DUBs with arginine substitutions should first be tested *in vitro* to ensure that this mutation indeed interferes with ubiquitin binding. Adopting this practice can mitigate spurious or dominant negative effects and ensure that any observed phenotypes or changes are due to loss of DUB activity alone rather than an increase in affinity for ubiquitin.

## Materials and Methods

### Cloning, protein expression, and purification

Rosetta 2(DE3) pLysS cells (EMD Millipore, Merck KGaA, Darmstadt, Germany) were transformed with three plasmids encoding (i) Ubp8$^{WT}$, Ubp8$^{C146A}$, Ubp8$^{C146S}$, or Ubp8$^{C146R}$ (pET-32a, EMD Millipore), (ii) Sus1 (pRSF-1, EMD Millipore), and (iii) Sgf73$^{(1-96)}$ (pCDFDuet-1 MCSII, EMD Millipore) which was cloned into the same vector as Sgf11 (pCDFDuet-1 MCSI, EMD Millipore). All versions of the DUBm complex were co-expressed and purified using the previously reported protocol for the expression and purification of wild-type DUBm [27]. Untagged ubiquitin (pET3a) was expressed in Rosetta 2 cells and, after lysis, was treated with 1% v/v perchloric acid to precipitate cellular proteins. The supernatant, containing ubiquitin, was dialyzed overnight into 50 mM sodium acetate pH 4.5, then run on a HiTrap SP column, and eluted over a 0–600 mM NaCl gradient in 50 mM sodium acetate pH 4.5. Pure fractions were pooled and buffer exchanged by gel filtration on a HiLoad S75 column into 20 mM HEPES pH 7.5, 50 mM NaCl, and 1 mM DTT.

USP4 (8–925) wild-type and C311A mutant were expressed and purified as in [32].

pOPINK-OTUD1 catalytic domain (residues 287–435) was transformed into *Escherichia coli* Rosetta 2 pLysS (Novagen) and grown to OD 0.6 followed by induction with 0.2 mM IPTG overnight at 20°C. Cells were lysed by sonication in buffer A (50 mM Tris,

50 mM NaCl, 5 mM DTT, pH 8.5), the lysate was cleared by centrifugation ($44,000 \times g$ for 30 min, 4°C) and subjected to a glutathione resin (GE Healthcare). The resin was washed with cold buffer B (50 mM Tris, 500 mM NaCl, 5 mM DTT, pH 8.5) and subsequently with cold buffer A. The GST-tag was removed by overnight incubation at 4°C with GST-tagged 3C Precision protease in buffer A. Eluted protein was further purified by anion-exchange chromatography and gel filtration in buffer A.

**Fluorescence polarization assays**

All USP4 pre-steady-state and equilibrium fluorescence polarization assays were performed as described in [32]. Ubiquitin conjugated to lysine-glycine and to a SMAD4-derived peptide [32] was a gift of Huib Ovaa.

Binding assays for OTUD1 interactions with Lys63-linked chains were performed using Lys63-linked diUb that was fluorescently labeled by a FlAsH-tag on the proximal ubiquitin (*21*). For this, diUb chains were diluted to 80 nM in FlAsH buffer (50 mM Tris, 50 mM NaCl, 0.1% β-mercaptoethanol, pH 7.6), and OTUD1$^{C320A}$ and OTUD1$^{C320R}$ were serially diluted in FlAsH buffer to the indicated concentration range. 10 µl of fluorescent diUb was mixed with equal volume of OTUD1$^{C320A}$ and OTUD1$^{C320R}$ at different concentrations and incubated in room temperature for 1 h before measurement. Fluorescence polarization was measured in 384-well format employing a Pherastar FS plate reader, using a fluorescence polarization module with excitation and emission wavelengths at 485 and 520 nm, respectively. A control was used for either linear di- or triUb molecules where 10 µl of FlAsH buffer was added instead. This control was also used for the normalization of anisotropy reading. All binding assays were performed in triplicate.

**Ubiquitin-AMC hydrolysis assay**

Assays were conducted in 384-well black polystyrene micro-plates at 30°C in a POLARstar Omega plate reader (BMG Labtech, Cary, NC) using an excitation wavelength of 385 nm and emission wavelength of 460 nm. Reactions were performed in DUBm assay buffer containing 50 mM HEPES, pH 7.6, 150 mM NaCl, 5 µM ZnCl$_2$, 5 mM dithiothreitol (DTT), and 7.5% DMSO. The wild-type DUBm and Ubp8 mutant complexes were held at a concentration of 125 nM. Ubiquitin-AMC (Boston Biochem, Cambridge, MA) was diluted into assay buffer and incubated at 30°C for 10 min inside the plate reader. 3 µl of recombinant DUBm was also pre-incubated at 30°C for 10 min before mixing with diluted ubiquitin-AMC buffer to a total volume of 30 µl. The release of AMC was followed at 460 nm, and the first 0–60 s of data was used to fit initial rate.

**Isothermal titration calorimetry (ITC)**

Isothermal titration calorimetry measurements were performed using a Microcal (Amherst, MA) ITC200 calorimeter at 25°C. DUBm wild-type, Ubp8 mutant complexes, K48-linked diubiquitin and ubiquitin samples were buffered with 20 mM HEPES, pH 7, 150 mM NaCl, 5 nM ZnCl$_2$, and 0.5 mM Tris (2-Carboxyethyl) phosphine hydrochloride (TCEP) and thoroughly degassed before use. The protein concentrations were determined by

amino acid analysis. The sample cell (0.22 ml) contained either 30 µM DUBm-Ubp8$^{WT}$ or DUBm Ubp8 mutant. A total of 20 injections of 40 µl of 0.3 mM K48-linked diubiquitin or monoubiquitin were carried out at 180 s intervals. The heat generated due to dilution of the titrants was subtracted for baseline correction. The baseline-corrected data were analyzed with Microcal Origin Ver. 7.0 software. All experiments were duplicated.

**Protein crystallization**

Protein crystals were grown from a complex of 7 mg/ml DUBm-Ubp8$^{C146A}$ and 1.8 mg/ml ubiquitin that was incubated on ice for 30 min prior to screening. Complex crystals were grown by hanging drop vapor diffusion using a 1:1 ratio of protein to mother liquor. Crystals grew within 2 days at 20°C in 17% PEG3350, 0.1 M HEPES pH 7.0, and 0.1 M ammonium sulfate. Crystals were looped and cryoprotected by stepwise incubation in mother liquor containing increasing concentrations of PEG3350 (17–33%), then flash-frozen in liquid nitrogen.

**Data collection, structure determination, and refinement**

X-ray diffraction data for the DUBm-Ubp8$^{C146A}$ and ubiquitin complex were collected at Stanford Synchrotron Radiation Lightsource beamline BL12-2. Data were collected on a Pilatus detector using a 10 µm beam at 50% transmission taking 1 s exposures with 0.25° oscillations over 180°. During data collection, the crystal rotated out of the beam, therefore frames 200–300 out of 720 total frames were discarded during data reduction and scaling. Data reduction, scaling, and merging were done in XDS and Aimless [44]). A 2.1 Å structure was determined by molecular replacement in Phaser (Phenix) using the coordinates of the wild-type DUBm bound to ubiquitin aldehyde (PDB ID: 3MHS) as the search model [27,45]. The structure was refined in PHENIX and Coot was used for manual model building [45,46]. Data collection and refinement statistics are shown in Table 1. PyMOL Version 1.5.0.4 (Schrödinger, LLC) was used to generate all structure figures. Coordinates have been deposited in the Protein Data Bank with ID 6AQR.

**Immunoblotting and immunoprecipitation**

Full-length, N-terminally HA-tagged OTUD1 constructs (WT, C320A, C320R) were cloned into pcDNA3.1 vector and transiently expressed in HEK293 cells. Two days after, transfection cells were washed twice in cold PBS and scraped on ice in lysis buffer (50 mM Tris–HCl pH 7.5, 150 mM NaCl, 1% NP-40) containing protease inhibitor cocktail (Roche) and 20 mM N-ethylmaleimide, and incubated for 30 min on ice. Samples were subsequently cleared by centrifugation for 10 min at $16,000 \times g$ at 4°C. For immunoblotting, equivalent amounts of cell lysates were separated on 4–12% Bis-Tris gel, transferred to nitrocellulose membranes, and incubated with the indicated antibodies. The signal was visualized with Pierce ECL Western Blotting Substrate (Thermo Scientific) and exposed on X-ray film.

Full-length USP14 (WT, C114A, C114R) and N-terminally Flagtagged PSMD4 constructs were cloned into pcDNA3.1 vector and

transiently expressed in HEK293 cells. Two days after, transfection cells were washed twice in cold PBS and scraped on ice in lysis buffer (50 mM Tris–HCl pH 7.5, 150 mM NaCl, 1% NP-40) containing protease inhibitor cocktail (Roche) and 20 mM N-ethylmaleimide, and incubated for 30 min on ice. Samples were subsequently cleared by centrifugation for 10 min at $16,000 \times g$ at 4°C. The cleared lysates were incubated with anti-Flag resin (Sigma-Aldrich) overnight at 4°C, subsequently washed four times with cold lysis buffer and eluted with Flag peptide. Eluted fractions were separated on 4–12% Bis-Tris gel, transferred onto PVDF membranes, and incubated with the indicated antibodies. The signal was visualized with Pierce ECL Western Blotting Substrate (Thermo Scientific) and exposed on X-ray film.

### Antibodies

The following antibodies were used in immunoblotting or immuno-precipitation:

Ub K63, rabbit monoclonal Ab, clone Apu3, Millipore (cat # 05-1308); HA, rat monoclonal Ab, clone 3F10, Roche (cat # 11 867 423 001); GAPDH, mouse monoclonal Ab, clone GAPDH-71.1, Sigma-Aldrich (cat # G8795); Ub, mouse monoclonal Ab, clone P4D1, Santa Cruz (cat # sc-8017); Flag, mouse monoclonal Ab, clone M2, Sigma-Aldrich (cat # F1804); USP14, rabbit monoclonal Ab, clone D8Q6S, Cell Signaling Technology (cat # 11931).

## Data availability

Coordinates and amplitudes have been deposited in the Protein Data Bank with accession code, 6AQR.

**Expanded View** for this article is available online.

## Acknowledgements

We would like to thank Tycho E.T. Mevissen for help with OTU biophysics. Supported by grants GM095822 and GM109102 from the National Institute of General Medical Sciences (C.W.). The work in the T.K.S. lab has been supported by the European Research Council (249997). The D.K. lab is supported by the Medical Research Council (U105192732), the European Research Council (309756, 724804), the Michael J. Fox Foundation and the Lister Institute for Preventive Medicine. The M.S. lab is supported by the Institutional Development Plan of University of Ostrava and The Ministry of Education, Youth and Sports (IRP03_2018-2020). M.C. and M.S. were supported by EMBO Long Term Fellowships. Use of the Stanford Synchrotron Radiation Lightsource, SLAC National Accelerator Laboratory, is supported by the U.S. Department of Energy, Office of Science, Office of Basic Energy Sciences under Contract No. DE-AC02-76SF00515. The SSRL Structural Molecular Biology Program is supported by the DOE Office of Biological and Environmental Research, and by the National Institutes of Health, National Institute of General Medical Sciences (including P41GM103393).

## Author contributions

Binding and structural studies of the SAGA DUB module were carried out by MEM, MTM, and MY, with input and guidance from CW. Studies of USP4 were carried out by MC, with input and guidance from TKS. Biophysical studies on OTUD1 were carried out by MS under guidance of DK. Cell biological studies on OTUD1 and USP14 were designed by MS and carried out by KG and MS.

MEM and CW wrote the initial manuscript, with significant contributions from DK, TKS, and MS. All authors edited the manuscript.

## Conflict of interest

The authors declare that they have no conflict of interest.

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
