## [Review Process File · EMBO Reports]

Active site alanine mutations convert deubiquitinases into high affinity ubiquitin-binding proteins

Marie E Morrow, Michael T Morgan, Marcello Celrici, Katerina Growkova, Ming Yan, David Komander, Titia K Sixma, Michal Simicek, Cynthia Wolberger

Review timeline:

Submission date:	20 December 2017
Editorial Decision:	30 January 2018
Revision received:	22 June 2018
Editorial Decision:	12 July 2018
Revision received:	24 July 2018
Accepted:	31 July 2018

Transaction Report:

1st Editorial Decision

30 January 2018

Thank you for the submission of your research manuscript to our journal. I apologize for the delay in handling your manuscript but we have only recently received the final referee report. Please find the full set of referee reports copied below.

As you will see, the referees acknowledge that the manuscript reports important information for the community. However, the referees also raise concerns regarding the novelty of the findings and they indicate that further work will be required to substantiate the findings but also to generalize the observed effects of a C-to-A mutation. Referee 3 considers it important to provide a new example how such a mutated DUB affects cellular function, a point that was supported by all referees upon further discussion. Referee 2 points out that the DUB expression levels should be taken into account when reporting on affinity and the affinity measurements should be done in a manner that eliminates potential artefacts caused by the fluorescent tags.

Given these constructive comments, we would like to invite you to revise your manuscript with the understanding that the referee concerns must be fully addressed and their suggestions as outlined above and in their reports taken on board. Please address all referee concerns in a complete point-by-point response. Acceptance of the manuscript will depend on a positive outcome of a second round of review. It is EMBO reports policy to allow a single round of revision only and acceptance or rejection of the manuscript will therefore depend on the completeness of your responses included in the next, final version of the manuscript.

Revised manuscripts should be submitted within three months of a request for revision; they will otherwise be treated as new submissions. Please contact us if a 3-months time frame is not sufficient for the revisions so that we can discuss the revisions further.

Supplementary/additional data: The Expanded View format, which will be displayed in the main HTML of the paper in a collapsible format, has replaced the Supplementary information. You can submit up to 5 images as Expanded View. Please follow the nomenclature Figure EV1, Figure EV2

etc. The figure legend for these should be included in the main manuscript document file in a section called Expanded View Figure Legends after the main Figure Legends section. Additional Supplementary material should be supplied as a single pdf labeled Appendix. The Appendix includes a table of content on the first page, all figures and their legends. Please follow the nomenclature Appendix Figure Sx throughout the text and also label the figures according to this nomenclature. For more details please refer to our guide to authors.

Regarding data quantification, can you please specify the name of the statistical test used to generate error bars and P values, the number (n) of independent experiments underlying each data point (not replicate measures of one sample), and the test used to calculate p-values in each figure legend? Discussion of statistical methodology can be reported in the materials and methods section, but figure legends should contain a basic description of n, P and the test applied. Please also include scale bars in all microscopy images.

We now strongly encourage the publication of original source data with the aim of making primary data more accessible and transparent to the reader. The source data will be published in a separate source data file online along with the accepted manuscript and will be linked to the relevant figure. If you would like to use this opportunity, please submit the source data (for example scans of entire gels or blots, data points of graphs in an excel sheet, additional images, etc.) of your key experiments together with the revised manuscript. Please include size markers for scans of entire gels, label the scans with figure and panel number, and send one PDF file per figure.

- a complete author checklist, which you can download from our author guidelines (<http://embor.embopress.org/authorguide#revision>). Please insert page numbers in the checklist to indicate where the requested information can be found.
 - a letter detailing your responses to the referee comments in Word format (.doc)
 - a Microsoft Word file (.doc) of the revised manuscript text
 - editable TIFF or EPS-formatted figure files in high resolution
- (In order to avoid delays later in the publication process, please check our figure guidelines before preparing the figures for your manuscript:
http://www.embopress.org/sites/default/files/EMBOPress_Figure_Guidelines_061115.pdf)
- a separate PDF file of any Supplementary information (in its final format)
 - all corresponding authors are required to provide an ORCID ID for their name. Please find instructions on how to link your ORCID ID to your account in our manuscript tracking system in our Author guidelines (<http://embor.embopress.org/authorguide>).

As part of the EMBO publication's Transparent Editorial Process, EMBO reports publishes online a Review Process File to accompany accepted manuscripts. This File will be published in conjunction with your paper and will include the referee reports, your point-by-point response and all pertinent correspondence relating to the manuscript.

I look forward to seeing a revised version of your manuscript when it is ready. Please let me know if you have questions or comments regarding the revision.

REFeree REPORTS

Referee #1:

SUMMARY

1 Does this manuscript report a single key finding? YES

Morrow et al. report that C-to-A substitutions of the catalytic residue of deubiquitylases converts these enzymes into avid ubiquitin-binding domains, which can cause artifacts when overexpressed in cells.

2 Is the reported work of significance (YES), or does it describe a confirmatory finding or one that has already been documented using other methods or in other organisms etc (NO)?

NO, see details below

3 Is it of general interest to the molecular biology community? YES

DUBs are important and widely studied enzymes and many researchers have been using C-to-A substitutions in cellular assays.

4 Is the single major finding robustly documented using independent lines of experimental evidence (YES), or is it really just a preliminary report requiring significant further data to become convincing, and thus more suited to a longerformat article (NO)?

YES, but more experimentation would be required to further strengthen their claims.

In this report, Morrow et al. study the influence of active site mutations of the deubiquitylation enzymes (DUBs) Ubp8 (in context of the SAGA DUB module), USP4, and OTUD1 on their DUB activity and ubiquitin binding properties. Using Ub-AMC assays and isothermal thermal calorimetry the authors find that while mutation of the catalytic C residue to S, R, or A in Ubp8 results in DUB inactivation, only C-to-A substitution increases binding of Ubp8 to mono- and di-ubiquitin. They provide a molecular explanation for this increase in affinity by solving the crystal structure of the SAGA DUB module containing Ubp8C146A bound to free ubiquitin, which shows that the C-terminal part of ubiquitin is bound in a manner that would result in a steric clash with the sulfhydryl group of the active site C residue. The authors then go on and show that analogous C-to-A substitutions in USP4 and OTUD1 result in similar affinity increases for free ubiquitin and K63-linked ubiquitin, respectively. Finally, Morrow et al. show that ectopic expression of OTUD1 C320A (but not WT or C320R) results in stabilization of K63-linked chains in HEK 293T cells.

The manuscript is well written and the experiments are of high quality and they support the authors main conclusion that active site C-to-A mutations in DUBs convert these enzymes into avid ubiquitin binding domains that when expressed in cells can result in off-target effects not related to inactivation of the DUB activity. DUBs are enzymes that have important functions in a vast variety of different biological processes and are studied by a broad scientific audience. Therefore, this is an important finding for the molecular biology community (who has been using C-to-A mutants in many ectopic expression studies) and has not been discussed as such in the literature.

However, I am concerned about the novelty of the findings. As acknowledged by the authors, it was already known that similar C-to-A substitution in OTUD7B (Cezanne) results in increased ubiquitin binding (Mevisse et al., Nature, 2016, Extended Data Figure 5). In addition, there are numerous studies that have successfully used the C-to-S mutation in DUBs (e.g. for USP7 - Li et al., Nature, 2002, USP25 - Zhong et al., Nat Immunol, 2012, etc.) suggesting that these C-to-S mutants are catalytically dead or their catalytic activity is negligible (as also demonstrated by the authors for Ubp8 in Supplementary Figure 1). This takes away from the novelty of the author's recommendation to use C-to-R mutations.

Specific points:

1. The authors provide experimental evidence for only USPs and OTUs, but claim that their findings can be expanded to all DUBs with catalytic cysteine residues. I think this is reasonable assumption, but their claim would be strengthened by providing more evidence. Ideally, this could be done by making C-to-A/R mutations in representative members of the UCHL, MJD, and MINDY family and measure affinities using their established FP assays. At the very least, they could show an overall

similar architecture of the catalytic pocket of all five families of DUBs by aligning available structures.

2. Figure 5 is the only figure that demonstrates that ectopic expression of a C-to-A DUB mutant (OTUD1) has off-target effects in cells. Again, given the existing literature, I think it is reasonable to assume that other C-to-A DUB mutants will have similar effects, but the report would be strengthened by experimentally showing this for at least another DUB family member. E.g.: what is the effect of overexpression of USP4 C311A versus C311R on cellular ubiquitin pools? Or what is the effect of overexpression of C-A/R mutants of proteasome-associated DUBs (USP14 or UCH37) on K48-linked ubiquitin levels or protein turnover?

3. While I do agree with the reasoning and statement on page 4 line 9 that expression of tight-binding DUB mutants will likely result in off-target effects, the example of A20 does not support this notion. First, I have a hard time conceptualizing how off-target effects of the tighter binding C-to-A A20 mutant should cause the absence of an expected phenotype in TNF α signaling in the knock in mice (De et al., EMBO Rep 2014). Second, and more importantly, C-to-A A20 knock in mice generated by a different group showed the expected phenotype in TNF α signaling (Wertz et al., Nature, 2015, Extended Data Figure 4). Therefore, this example should be removed from the text.

Referee #2:

This manuscript by Morrow et al. shows that, for several deubiquitinating enzymes (DUBs) of the USP or OTU families, substitution of Ala for the active-site Cys dramatically strengthens their binding to ubiquitin or ubiquitin derivatives. The measured affinity increases ranged from 10 to 150-fold. Because active-site C-to-A mutations are used widely to inactivate DUBs and evaluate their functions, the authors suggest that the tighter substrate binding by the C-to-A mutated enzymes could have unintended effects and might introduce artifacts when the altered DUBs are expressed. An example with OTUD1(C320A) expressed in HEK293 cells (Fig. 5) was used to illustrate this point.

Experiments were done to evaluate changes in binding affinities upon substitutions of Ser, Ala, or Arg in different DUBs for the active-site Cys. Also, a high-resolution crystal structure was determined for monoubiquitin bound to Ubp8(C146A) of the SAGA DUB module. The results provide a possible explanation for enhanced ubiquitin binding relative to the wild type: because the C-to-A replacement makes the active site less sterically constrained, it can better accommodate ubiquitin's C-terminal carboxylate. For Ubp8(C146A), two H-bonds to ubiquitin G76 appear to stabilize ubiquitin; these H-bonds would not form in wild-type Ubp8. Because the active-site structures of USP and OTU DUBs are highly conserved, the C-to-A replacements may have similar consequences. DUB inactivation by substitution of the Cys with Arg, which would not promote tight ubiquitin binding, was recommended as an alternative to Ala.

The authors make a convincing case that C-to-A substitution in most Cys-based DUBs is likely to increase the affinity for ubiquitin or ubiquitin conjugates. However, as detailed below, the conclusions about the magnitudes of those affinity changes and the likely structural basis for the effects need additional supporting evidence. Moreover, the significance of the authors' observations regarding enhanced ubiquitin binding by C-to-A variants - i.e., that expression of such a DUB variant can introduce artifacts relative to other substitutions (e.g., C-to-R) - needs qualification and further discussion.

1. Results of binding assays using various modified forms of ubiquitin (e.g., TAMRA-ubiquitin with Usp4, and FLAsH-diubiquitin with OTUD1) were reported and discussed as if the K_d values were determined for unmodified ubiquitin (e.g., Fig. 3 legend and p. 7). That these bulky hydrophobic tags might have influenced the affinities cannot be ignored. Indeed, it is quite possible that interaction with the fluorophores contributed to the enhanced binding by the C-to-A DUBs, especially with USP4 where a 150-fold enhancement was observed. Affinities of unmodified (di)ubiquitin needs to be determined; one approach would be to extend the fluorescence-based assays to measure (di)ubiquitin affinities as K_i values in competition experiments

2. Enhanced binding by Ubp8(C146A) and, by inference, other C-to-A DUBs is hypothesized to be due in part to hydrogen bonds made to ubiquitin's G76 carboxylate. This is a key feature of the paper. It should be tested biochemically. For example, do wt and C146 Ubp8 bind similarly to a truncated (i.e., delta-G76) ubiquitin?
3. In Fig. 4, binding by wild-type OTUD1 should be included for comparison.
4. Given the rather modest affinity of OTUD1(C320A) for K63-diubiquitin (from Fig. 4, K_d appears to be ~25 uM), the dramatic intracellular enrichment of K63-polyubiquitin seen in Fig 5 is surprising. It's difficult to imagine how that could occur unless the OTUD1(C320A) was massively overexpressed. What was the expression level? Especially as the potential for artifacts by C-to-A DUBs is a major theme in the paper, the authors should discuss the related issues of affinity and expression level.
5. Figs. 4 and 5 should be combined into a single figure.
6. The term "avid" in biochemistry usually refers to multivalent binding interactions. Because that's not the case with the binding reactions in this study, a different word should be used in the title to avoid confusion.
7. In Fig. S3, the magnitudes of the fast and slow phases fit to the kinetic data in panels A and B should be reported. Are the kinetics consistent with the K_d values determined from binding at equilibrium?
8. Finally, the authors might want to look to see if active-site C-to-A DUB variants occur naturally. In analogy with pseudo-kinases and, for ubiquitin, the UEV/Mms2 pseudo-E2 enzymes, one could imagine that ubiquitin binding proteins may have evolved from inactive pseudo-DUBs.

Referee #3:

Nature created catalytic inactive enzymes as substrate binding proteins. For example, the plant seed storage proteins concanavalin B and narbonin are catalytically inactive chitinases that instead function to bind oligomeric sugars rather than cleaving them. Similarly, UEV is E2 variant that lacks the catalytic cysteine and functions as a Ub-binding domain rather than actually conjugating Ub to protein targets.

Here, Wolberger and co-workers show that catalytic inactive DUBs designed by researchers, may function to bind mono and poly-Ub especially where the mutation is Cys>Ala. Using biophysical measurements, they demonstrated a significant (up to 150 folds) increased affinity between such mutated DUBs and Ub or diUb. They provide a simple structural explanation for the phenomenon and demonstrate that the alternative substitution C>R is not only catalytically inactive, but also (probably because it repels the Ub), does not show increased affinity.

The authors start the discussion saying that their finding is surprising. However, as mentioned above substrate binding of catalytically dead mutant enzymes is the nature way to design ligand binding proteins. Moreover, structural biologists routinely take this approach to capture enzyme:substrate complex.

Nevertheless, the report is kind of a WARNING CALL to the community to not use such mutants in cells due to their possible effect. The authors claim that overexpressed C>A DUB, present a risk that cellular consequences ascribed to a lack of catalytic activity in a particular DUB may instead function as scavenger of polyUb or to affect the evaluability of free Ub to the Ub-systems. While they demonstrated the capability of OTUD1 C320A to stabilize K63 polyUb, the manuscript falls short to actually address the claimed warning.

If indeed some of these C>A mutants affect cellular process due to tight binding of Ub or Ub chains (which I certainly believe) the outcome conclusion is that overexpression of many other Ub-receptor may present similar effect. Similarly, overexpression of catalytically dead ubiquitylating enzymes (E1s, E2s and E3s) containing UBDs, would potentially function in the same manner. But this is not new. There are many examples showing the effect of over expression in cells including Ub-receptors. I will refer just to one example here: the effect of changing the concentration of Ub-receptors including Rpn10, Rad23 and Dsk2 on embryonic development was demonstrated by Lipinszki et al JCS 2009.

Taking together, I do not see any conceptual novelty in this manuscript, but only an alert to the

community.

Hence, without showing a new example how such mutated DUB affect cellular function I do not think the manuscript suits the level of EMBO Report.

On the other hand, due to the large size of the Ub community I definitely see the importance of this warning alert. I therefore recommend that if the authors can give such an example and the manuscript will eventually be published in EMBO Rep. it should be more general (at least in the text) to include other Ub enzymes and Ub-receptors.

It worth noting that the experiments were performed in elegant manner. The presentations are clear, figures are self-explanatory as well as the text. The crystallographic data seems to be excellent. It is nice to see that although ~1/3 of the collected images were of uncentered crystal in the beam, the final product (as judged by the electron density map and the statistics) is very convincing.

Technical comments:

1. The titration isotherms in figure 1 are very nosy. Did the DUB and Ub (ligand) were dialyzed simultaneously in the same bucket? I would like to clarify that I DO NOT think the experiments should be redone, but only to learn the reason for the noisy data.

2. Figure 5. The title of the legend says in vivo, however, it was done in HEK293. Maybe say in cell line instead of in vivo.

1st Revision - authors' response

22 June 2018

RE: MANUSCRIPT EMBOR-2017-45680V1

RESPONSE TO REVIEWERS

REVIEWER 1

Comment: *“The authors provide experimental evidence for only USPs and OTUs, but claim that their findings can be expanded to all DUBs with catalytic cysteine residues. I think this is reasonable assumption, but their claim would be strengthened by providing more evidence. Ideally, this could be done by making C-to-A/R mutations in representative members of the UCHL, MJD, and MINDY family and measure affinities using their established FP assays. At the very least, they could show an overall similar architecture of the catalytic pocket of all five families of DUBs by aligning available structures.”*

Response: We agree and have aligned the available ubiquitin-bound structures for representatives of the UCH (UCHL1) and MJD (ataxin-3) families with our Ubp8 C146A active site. Due to the lack of structural conservation of the MINDY DUB family with other cysteine protease DUBs, it does not align to other cysteine proteases so we have shown the active site architecture of MINDY alone. However, as described below, we have found data in the literature demonstrating that the MINDY C-to-A mutant similarly increases the affinity of MINDY for ubiquitin. This increase in affinity can be seen in a figure but was not commented on in the paper; we therefore point it out in our manuscript.

Comment: *“Figure 5 is the only figure that demonstrates that ectopic expression of a C-to-A DUB mutant (OTUD1) has off-target effects in cells. Again, given the existing literature, I think it is reasonable to assume that other C-to-A DUB mutants will have similar effects, but the report would be strengthened by experimentally showing this for at least another DUB family member. E.g.: what is the effect of overexpression of USP4 C311A versus C311R on cellular ubiquitin pools? Or what is the effect of overexpression of C-A/R mutants of proteasome-associated DUBs (USP14 or UCH37) on K48-linked ubiquitin levels or protein turnover?”*

Response: All three reviewers were concerned about a lack of novel biology demonstrating our hypothesis, so this experiment was done with USP14 to extend the evidence for our warning to the community. We ectopically expressed USP14 wild-type, C114A, and C114R and immunoprecipitated proteasomes via PSMD4/Rpn10. Polyubiquitin levels were enriched in USP14 C114A cells.

In addition, we cite another example of spurious effects that can be attributed to a C-to-A mutant in the literature. In Keusekotten et al. (*Cell* **153**: 1312-1326), An OTULIN C to A mutation gave

rise to high levels of protected linear ubiquitin, whereas an OTULIN mutant defective in ubiquitin binding showed much lower levels of linear ubiquitin.

Comment: “While I do agree with the reasoning and statement on page 4 line 9 that expression of tight-binding DUB mutants will likely result in off-target effects, the example of A20 does not support this notion. First, I have a hard time conceptualizing how off-target effects of the tighter binding C-to-A A20 mutant should cause the absence of an expected phenotype in TNF α signaling in the knock in mice (De et al., EMBO Rep 2014). Second, and more importantly, C-to-A A20 knock in mice generated by a different group showed the expected phenotype in TNF α signaling (Wertz et al., Nature, 2015, Extended Data Figure 4). Therefore, this example should be removed from the text.”

Response: We have removed this example from the text.

REVIEWER 2

Comment: “Results of binding assays using various modified forms of ubiquitin (e.g., TAMRAubiquitin with Usp4, and FLAsH-diubiquitin with OTUD1) were reported and discussed as if the K_d values were determined for unmodified ubiquitin (e.g., Fig. 3 legend and p. 7). That these bulky hydrophobic tags might have influenced the affinities cannot be ignored. Indeed, it is quite possible that interaction with the fluorophores contributed to the enhanced binding by the C-to-A DUBs, especially with USP4 where a 150-fold enhancement was observed. Affinities of unmodified (di)ubiquitin needs to be determined; one approach would be to extend the fluorescence-based assays to measure (di)ubiquitin affinities as K_i values in competition experiments”

Response: We would argue that these measurements are intended to show relative differences between the C->A mutants and either wild-type or C->R, not absolute K_d values. The same fluorescent-labeled ubiquitin substrate is used for both wild-type and C->A and the data show significant differences in binding affinity. The DUB field regularly uses fluorophore-modified ubiquitin substrates for measuring catalytic activity and binding. In the experiments reported in our study, the fluorophore does not interact with DUB active site residues. Both the TAMRA and FLAsH- labels are installed on the N-terminus of ubiquitin, which is not a binding interface for any of the DUBs used in this study. In the case of Ubp8, direct measurements with unlabeled ubiquitin were used, showing the same trend.

Comment: “Enhanced binding by Ubp8(C146A) and, by inference, other C-to-A DUBs is hypothesized to be due in part to hydrogen bonds made to ubiquitin's G76 carboxylate. This is a key feature of the paper. It should be tested biochemically. For example, do wt and C146 Ubp8 bind similarly to a truncated (i.e., delta-G76) ubiquitin?”

Response: The reviewers raise an interesting idea and suggest testing binding of G75 Ub to Ubp8 wt and C146A. However, using G75 ubiquitin would introduce an extra negatively charged carboxylate, thus interrupting the canonical hydrogen bonding of DUB active site clefts to the backbone of ubiquitin's C-terminus. In the DUBm Ubp8C146A – Ub structure presented in the paper, the backbone amide from the peptide bond between G75 and G76 of ubiquitin hydrogen bonds to the carboxyl group of G426 of Ubp8. If G76 were removed, this backbone amide would not be able to hydrogen bond to G426 and a repelling negative charge would be introduced, likely resulting in overall weaker ubiquitin binding. We therefore do not think that G75 ubiquitin would mimic what the reviewer hopes to see.

Comment: “Given the rather modest affinity of OTUD1(C320A) for K63-diubiquitin (from Fig. 4, K_d appears to be ~25 μ M), the dramatic intracellular enrichment of K63-polyubiquitin seen in Fig 5 is surprising. It's difficult to imagine how that could occur unless the OTUD1(C320A) was massively overexpressed. What was the expression level? Especially as the potential for artifacts by C-to-A DUBs is a major theme in the paper, the authors should discuss the related issues of affinity and expression level.”

Response: The goal of this paper is to warn cell biologists, who frequently over-express proteins to see their effects in the cell, not to use DUB C->A mutations because a dramatic enrichment may occur and would be undesired. We intentionally overexpressed these DUBs to show what artifacts are possible in cell studies using these mutants. However, during some

ubiquitin signaling events, it is possible that even a DUB with modest affinity for polyubiquitin could be presented with a high local concentration of ubiquitin species. During cellular stress events such as DNA damage, autophagy, inflammatory signaling, and immune responses, ubiquitin signals accumulate in very high local concentrations.

Even DUBs that bind ubiquitin weakly have a temporarily high concentration of substrate during cellular stress responses. Our *in vitro* data indicates that C->A mutants release ubiquitin bound species with a slower off-rate, implying that these mutants may sequester ubiquitin species at detrimentally high concentrations during a signaling event.

Comment: *“In Fig. S3, the magnitudes of the fast and slow phases fit to the kinetic data in panels A and B should be reported. Are the kinetics consistent with the Kd values determined from binding at equilibrium?”*

Response: The off-rates for USP4 C to A mutant are reduced by ~100 fold with respect to the wild-type protein whereas on-rates are in the same range, consistent with a reduction in Kd of ~150 fold measured at equilibrium. Additional data to clarify this point has been added to Figure S3.

Comment: *“The term “avid” in biochemistry usually refers to multivalent binding interactions. Because that’s not the case with the binding reactions in this study, a different word should be used in the title to avoid confusion.”*

Response: We thank the reviewer for pointing this out. We have changed the title to reflect more accurately the biochemistry we are describing.

Comment: *“Finally, the authors might want to look to see if active-site C-to-A DUB variants occur naturally. In analogy with pseudo-kinases and, for ubiquitin, the UEV/Mms2 pseudo-E2 enzymes, one could imagine that ubiquitin binding proteins may have evolved from inactive pseudo-DUBs.”*

Response: We thank the reviewer for suggesting this interesting possibility. However, we are unaware of naturally occurring cysteine to alanine DUB variants. The inactive DUBs USP39 and FAM105A have classic USP/UBP and OTU domains, respectively but each has its catalytic cysteine residue substituted with an aspartic acid. An alignment of the USP domain of another inactive DUB, USP52/PAN2, with that of UBP8 indicates that instead of a catalytic cysteine, USP52 has a leucine substitution. To our knowledge, most of the disease-associated DUB mutations are SNPs or mutations rendering the DUB inactive by improper folding of the protein or premature stop codons leading to expression of a non-functional protein.

Minor corrections and concerns:

i) Figs. 4 and 5 should be combined into a single figure.

Response: The figures have been combined as suggested.

REVIEWER 3

Comment: *“Taking together, I do not see any conceptual novelty in this manuscript, but only an alert to the community. Hence, without showing a new example how such mutated DUB affect cellular function I do not think the manuscript suits the level of EMBO Report.”*

Response: As we addressed above in comments from Reviewer 1, we have added another example of the effects of ectopic expression of USP14 C114A on accumulation of polyubiquitin at the 26S proteasome, compared to wild-type and C114R. We also report a new crystal structure showing that the C-to-A mutation in Ubp8/SAGA DUB module allows new hydrogen bonding interactions and removes steric clashes that are predicted to occur in the wild type DUB. We hope that this addition of novel biology satisfies concerns by both Reviewer 1 and 3. As mentioned above, we also feel it is important to alert the ubiquitin community to the unintended consequences of these DUB mutations.

Minor corrections and concerns:

(i) “The titration isotherms in figure 1 are very noisy. Did the DUB and Ub (ligand) were dialyzed simultaneously in the same bucket? I would like to clarify that I DO NOT think the experiments should be redone, but only to learn the reason for the noisy data.”

Response: The DUB and Ub were dialyzed in the same container over multiple buffer exchanges. For some of these data, the heats of binding were very weak. When the data was

plotted, scales automatically adjust to the largest value, which visually enhances the thermogram noise.

(ii) “Figure 5. The title of the legend says *in vivo*, however, it was done in HEK293. Maybe say in cell line instead of *in vivo*.”

Response: We will change the text to clarify that these experiments were done in cell lines, not *in vivo*.

2nd Editorial Decision

12 July 2018

Thank you for the submission of your revised manuscript to our editorial offices. We have now received the reports from the referees that were asked to re-evaluate your study (you will find enclosed below). As you will see, the referees now support the publication of your manuscript in EMBO reports. However, referees #1 and #2 have some further suggestions to improve the paper that we ask you to address in a final revised version of the manuscript. As indicated by referee #3, adding data showing binding of other UB derivatives is not mandatory, but in case you have such data, we would ask you to add these to the final revised manuscript.

From the editorial side there are also a few things that need to be addressed. (Or some sentence along these lines....)

REFeree REPORT

Referee #1:

In this revised manuscript, Morrow et al. have provided additional data and evidence to strengthen their warning to the ubiquitin community that alanine-to-cysteine substitutions in deubiquitylases (DUBs) convert these enzymes into stronger ubiquitin binders, which can result in off-target effects when these proteins are ectopically expressed in cells.

My main concern was the limited amount of experimental evidence that this artifact is likely to occur with DUBs other than OTUD1. The authors have now provided additional experimental data for the proteasome-associated DUB USP14 (new Figure 5) as well as evidence from the literature for OTULIN (Keuskotten et al, Cell 2013). In addition, the authors show conservation of the active site architecture between USP DUBs (i.e. Ubp8) and members of the UCH and MJD classes (new Figure 6). Thus, I think the authors' conclusion that unwanted effects of cysteine to alanine substitutions will extend to many cysteine protease DUBs of different families is now sufficiently strengthened.

As I pointed out in my first review, these findings are not necessarily novel, but contain an important warning message for the ubiquitin community and should therefore be published in EMBO Reports.

Minor points:

1) In the rebuttal the authors also mention that MINDY C-to-A substitution leads to increased ubiquitin binding, but never explain their argument in detail. I assume the authors are referring to the study by Rehman et al, Mol Cell, 2016. The only experiment that I can find and that addresses ubiquitin binding of WT and C137A MINDY is Figure 4G, yet without a proper control (e.g. C137R as reference) this experiment does not support their claims. I could be missing something, so maybe the authors could explain in more detail, please?

Referee #2:

I am satisfied with the authors' responses to the reviewers' critiques (see, however, the two points elaborated below) and think that the work is now acceptable for publication.

One issue is really just a point of information for the authors regarding the idea that C-to-A substitutions might be employed to generate UBDs from "pseudo-DUB" proteins. In human and mouse USP52, an Ala residue (not Leu) is found at the position normally occupied by the active-site Cys, so USP52 might indeed function to bind ubiquitin. There's some modest support for that idea from Ron Hay and colleagues (Biochem J 2013) who had reported that immunoprecipitates of USP52 contained ubiquitin.

The more substantial issue is in regard to the authors' response to the suggestion that they provide biochemical evidence to support their claim that H-bond contacts to the G76 carboxylate stabilize ubiquitin binding in the C-to-A mutant (e.g., as depicted in Fig. 2B). Whereas I agree that UbG75, which was suggested originally, may not be the ideal binding ligand to test, I don't understand the authors' concern in their rebuttal that "...G75 ubiquitin would introduce an extra negatively charged carboxylate". In any case, binding to other Ub derivatives (e.g., UbG75-amide or ester) could be tried; however, much as I'd like to see those results, I don't consider those experiments to be a prerequisite to publication.

Referee #3:

The authors addressed the most critical comments and now the manuscript is ready for publication in EMBO Rep.

2nd Revision - authors' response

24 July 2018

Response to reviewers:

Referee #1:

In this revised manuscript, Morrow et al. have provided additional data and evidence to strengthen their warning to the ubiquitin community that alanine-to-cysteine substitutions in deubiquitylases (DUBs) convert these enzymes into stronger ubiquitin binders, which can result in off-target effects when these proteins are ectopically expressed in cells.

My main concern was the limited amount of experimental evidence that this artifact is likely to occur with DUBs other than OTUD1. The authors have now provided additional experimental data for the proteasome-associated DUB USP14 (new Figure 5) as well as evidence from the literature for OTULIN (Keuskotten et al, Cell 2013). In addition, the authors show conservation of the active site architecture between USP DUBs (i.e. Ubp8) and members of the UCH and MJD classes (new Figure 6). Thus, I think the authors conclusion that unwanted effects of cysteine to alanine substitutions will extend to many cysteine protease DUBs of different families is now sufficiently strengthened.

As I pointed out in my first review, these findings are not necessarily novel, but contain an important warning message for the ubiquitin community and should therefore be published in EMBO Reports.

Minor points:

1) In the rebuttal the authors also mention that MINDY C-to-A substitution leads to increased ubiquitin binding, but never explain their argument in detail. I assume the authors are referring to the study by Rehman et al, Mol Cell, 2016. The only experiment that I can find and that addresses ubiquitin binding of WT and C137A MINDY is Figure 4G, yet without a proper control (e.g. C137R as reference) this

experiment does not support their claims. I could be missing something, so maybe the authors could explain in more detail, please?

Response: We agree with the reviewer and had, in fact, removed this reference from the text but neglected to omit this from our response to reviewers. Apologies.

Referee #2:

I am satisfied with the authors' responses to the reviewers' critiques (see, however, the two points elaborated below) and think that the work is now acceptable for publication.

One issue is really just a point of information for the authors regarding the idea that C-to-A substitutions might be employed to generate UBDs from "pseudo-DUB" proteins. In human and mouse USP52, an Ala residue (not Leu) is found at the position normally occupied by the active-site Cys, so USP52 might indeed function to bind ubiquitin. There's some modest support for that idea from Ron Hay and colleagues (Biochem J 2013) who had reported that immunoprecipitates of USP52 contained ubiquitin.

Response: The reviewer makes an interesting point. USP52, however, appears to be a bit divergent based on sequence alignments and comparisons with the structure of the yeast homologue (PDB ID 4Q8G). This pseudo-DUB may therefore be sufficiently divergent that the analogy with other USP DUBs might not quite hold, although that remains to be seen. A recent report indicates that USP52 may have catalytic activity, but only when purified from insect cells that may retain post-translational modifications important for a catalytically-competent conformational change (Yang et al, Nat Communications 2018). In short, this example is sufficiently confounding that we chose not to include it.

The more substantial issue is in regard to the authors' response to the suggestion that they provide biochemical evidence to support their claim that H-bond contacts to the G76 carboxylate stabilize ubiquitin binding in the C-to-A mutant (e.g., as depicted in Fig. 2B). Whereas I agree that UbG75, which was suggested originally, may not be the ideal binding ligand to test, I don't understand the authors' concern in their rebuttal that "...G75 ubiquitin would introduce an extra negatively charged carboxylate". In any case, binding to other Ub derivatives (e.g., UbG75-amide or ester) could be tried; however, much as I'd like to see those results, I don't consider those experiments to be a prerequisite to publication.

Response: We thank the reviewer for suggesting this experiment. It would be interesting to determine whether truncated ubiquitin binds equally well to C146A Ubp8.

Referee #3:

The authors addressed the most critical comments and now the manuscript is ready for publication in EMBO Rep.

Corresponding Author Name: Cynthia Wolberger

Manuscript Number: EMBOR-2017-45680V2